# Protocol for the 'exploration of mental well-being, health-related quality of life, and psychosocial outcomes in youths on dialysis in New Zealand: A cross-sectional methods study'

Kavitha Jaganathan[1], Anna Serlachius[2], Chanel Prestidge[3‡], Fredric S. Doss[4,5‡], Mark R. Marshall[1,6]*

1 Department of Medicine, Faculty of Medical and Health Sciences, University of Auckland, Auckland, New Zealand, 2 Department of Psychological Medicine, Faculty of Medical and Health Sciences, University of Auckland, Auckland, New Zealand, 3 Department of Paediatric Nephrology, Starship Children's Health, Te Whatu Ora Health New Zealand Te Toka Tumai Auckland, Auckland, New Zealand, 4 Renal Physiology, Auckland City Hospital, Te Whatu Ora Health New Zealand Te Toka Tumai Auckland, Auckland, New Zealand, 5 Department of Science, Auckland University of Technology, Auckland, New Zealand 6 Department of Medicine, Waitākere Hospital, Te Whatu Ora Health New Zealand Waitematā, Auckland, New Zealand,

☯ These authors contributed equally to this work.
‡ These authors also contributed equally to this work.
* mark.marshall@waitematadhb.govt.nz

## Abstract

### Introduction

It is well accepted that young individuals on dialysis have poor clinical outcomes, although it is increasing apparent that they have similarly poor patient-centred ones. We study these latter outcomes in the population of youths undergoing dialysis in Aotearoa New Zealand (AoNZ). This study aims to report their sociodemographic and clinical characteristics, and explore their mental well-being and health-related quality of life, and how these outcomes are influenced by factors such as self-efficacy, physical activity, mood, and social anxiety.

### Methods

A cross-sectional survey will be conducted of youths aged 15–24 years on dialysis in AoNZ with eligible youths on dialysis across AoNZ, using validated psychometric instruments to assess daily activities, mental health, quality of life, social interactions, and self-efficacy. Open-ended questions within the survey will collect qualitative data. Clinical and demographic data will be established from clinical records and linked public health databases.

### Analysis

Quantitative data will be analyzed using descriptive statistics and structural equation modeling to explore associations between psychometric constructs, mental

**Data availability statement:** Authors agree to make data and materials supporting the results or analyses presented in their paper available upon reasonable request. It is up to the authors to determine whether a request is reasonable. Several factors will be considered in evaluating requests: • Legitimacy of scientific purpose: requests will be viewed more favorably if the intention is replication, meta-analysis, secondary analysis, or educational use. • Requester credentials: requests from researchers at recognized institutions with relevant expertise will be viewed more favorably. • Feasibility and burden: reasonable requests should not impose excessive work on the authors. • Ethical and legal constraints: certain data containing sensitive information have restrictions on sharing, as specified in data use agreements and ethics / IRB approvals. In particular, as a matter of national policy in Aotearoa New Zealand (AoNZ), there is a relative prohibition around sending Māori research data overseas consistent with the principles of Māori self-determination and control over data about their communities, rooted in Te Tiriti o Waitangi (the Treaty of Waitangi) and as specified in the (AoNZ) Data and Statistics Act 2022. Sharing of such data would require an evaluation from the national (AoNZ) Health and Disability Ethics Committee and as well as IRBs of all the participating hospitals, who will decide whether it is possible enforce AoNZ rules and norms pertaining to such data, including but not limited to: (1) whether the overseas jurisdictions have comparable data protection safeguards to AoNZ; (2) whether there will be culturally appropriate representation on overseas governance committees to ensure that the data are handled in culturally appropriate ways with a good understanding of the NZ Māori community that the data relate to. • Specificity: only specific requests for variables relevant to well-defined research question will be considered. • Timing: requests made shortly after publication will be considered whereas those made a substantial amount of time later may not be possible when authors have moved on or data storage has lapsed. • Reciprocity and collaboration: offers of collaboration or data sharing in return will be viewed more favorably than one-sided requests. Requests should be

well-being, and quality of life. Open-ended questions will be analysed using qualitative content analysis.

## Ethics and Dissemination

Informed consent will be sought from all participants. The research will adhere to data protection guidelines, ensuring that participants' rights and confidentiality are protected. Results will be disseminated through academic publications, conferences, and summary reports provided to participants and stakeholders.

## Introduction

Chronic kidney disease (CKD) is an important health issue of our time [1,2]. It is a major global cause of lost life-years and affects over 850 million individuals [3]. Behind these statistics, there lies profound personal suffering for individuals who reach kidney failure, experiences that are often hard for healthcare professionals to fully appreciate. Amongst the most vulnerable kidney failure populations are classified as youth. Those aged 15–24 years are generally only a small proportion of the total kidney failure population, constituting only 1–2% of the total kidney replacement therapy patient pool in Aotearoa New Zealand (AoNZ) [4]. Even with a small population, however, such patients are a tragedy of missed potential and reduced life-years lived. From a societal perspective, this demographic is associated with a high cumulative cost, both directly from their greater longevity relative to older patients, and also indirectly from lost opportunity cost.

Cumulative clinical experience and medical research shows that the health of all patients with kidney failure is critically dependent on them receiving a timely kidney transplant, since this allows a quantity and quality of life that is closer to that of healthy people, while those remaining on dialysis have unquestionably poorer longevity and patient-centred outcomes [5,6]. Nonetheless, a substantial number of youths have to remain on dialysis, either temporarily as a bridge to transplant or a permanent modality (Fig 1) [7]. The physical and mental health of these individuals is not as well studied as it is in their counterparts receiving a transplant. What data exist suggest poor development of exercise capacity and psychosocial health, often impairing later achievement of vocational, recreational and relationship goals [8–11]. Understanding psychometric causes of these associations might provide an opportunity for intervention to improve the longer-term health status of youths, and benefit society through their greater life participation and community engagement.

This research aims to investigate the unmet needs in physical and psychosocial health among AoNZ youths undergoing dialysis. From these data, we will model likely relationships between psychometric constructs, physical activity and mental health in this population. Using a survey, we aim to assess the entire youth population on dialysis in AoNZ. We anticipate our results to provide valuable insights into their specific needs and inform the development of targeted interventions to enhance their mental well-being and quality of life (QOL).

directed to the corresponding author and to
hdecs@health.govt.nz quoting ethics reference:
21/NTB/125.

**Funding:** University of Auckland Open Access
Grant as part of general PhD funding.

**Competing interests:** The authors have
declared that no competing interests exist.

## Materials and methods

### Study design and setting

This observational cross-sectional study will focus on youth on dialysis in AoNZ. Data will be collected without altering or intervening in treatment protocols, emphasizing real-world experiences and outcomes. Patient perceptions and beliefs will be assessed through a survey, supplemented by clinical and demographic data sourced from clinical records and linked public health databases. The study will be conducted in accordance with applicable aspects of Good Clinical Practice guideline [12]. Ethical approval has been granted by the national (NZ) Health and Disability Ethics Committee (21/NTB/125) as well as Institutional Review Boards (IRBs) of all the participating hospitals. All participants will provided informed consent. Patient recruitment started in 13 June 2022 and is expected to be completed during early December of 2025, and data analysis completed and results submitted for publication in early 2026.

### Patient and public partnership

To refine the study protocol, we have engaged both public and patients through an iterative process. Feedback has been collected from clinical practitioners and the University of Auckland Health Psychology Research Lab Group, including healthy individuals. The survey was piloted on five young people living with chronic kidney disease, and subtle adjustments made to order and layout. This approach has been adopted to ensure a patient-centered and practical design [13,14].

### Study population

**Inclusion criteria.** Eligible patients for the study are those aged between 15 and 24 year [15], inclusive, who are currently undergoing dialysis treatment in AoNZ, and able to provide informed consent or assent. Based on the Australian and New Zealand Dialysis and Transplant Registry (ANZDATA, www.anzdata.org.au), there are approximately 50 youths aged 15–24 currently receiving dialysis in Aotearoa New Zealand. We anticipate a recruitment rate of 50–60%, yielding an expected sample size of 25–30 participants.

**Exclusion criteria.** Exclusion criteria include patients posing logistic or safety risks while requiring interviewer assistance; those suffering from acute severe medical illness; those with severe functional difficulties or disabilities that are not appropriate for assessment by the survey instrument; those who are unable to give direct informed consent or assent; those outside the specified age range; and those not currently receiving dialysis treatment at the actual time of survey.

**Participant recruitment.** Invitations to participate will be made to patients by the University of Auckland (UoA) Research Team, after initial dissemination of information about the research study using letters, posters, texts and emails distributed by clinical staff within dialysis services. Patients agreeing to participate will be asked for their written informed consent and assigned an encrypted identifying code (eID) by local site investigators.

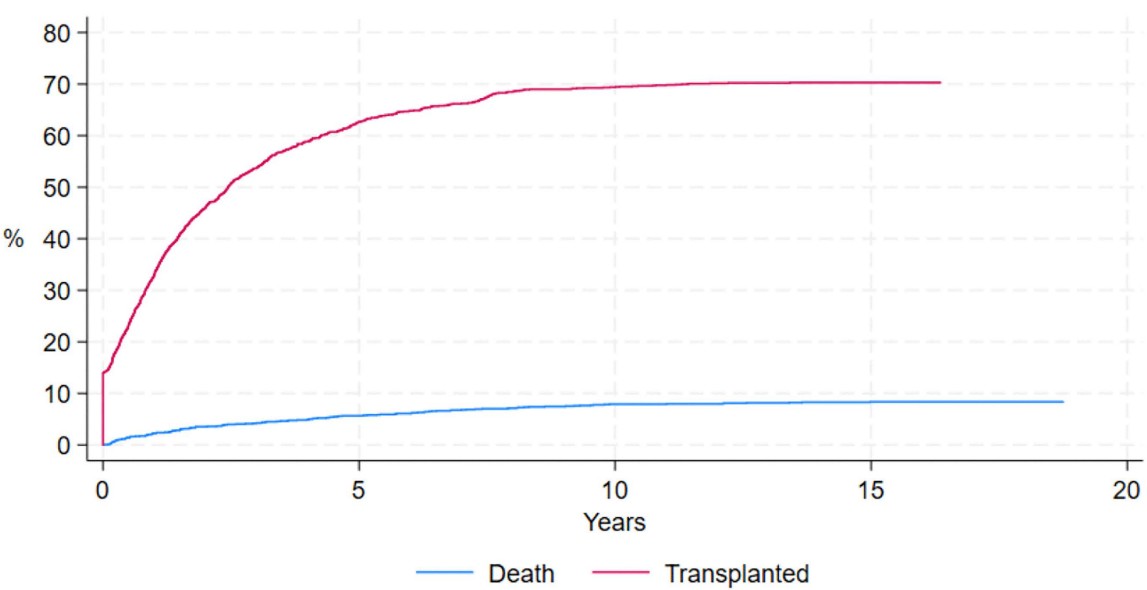

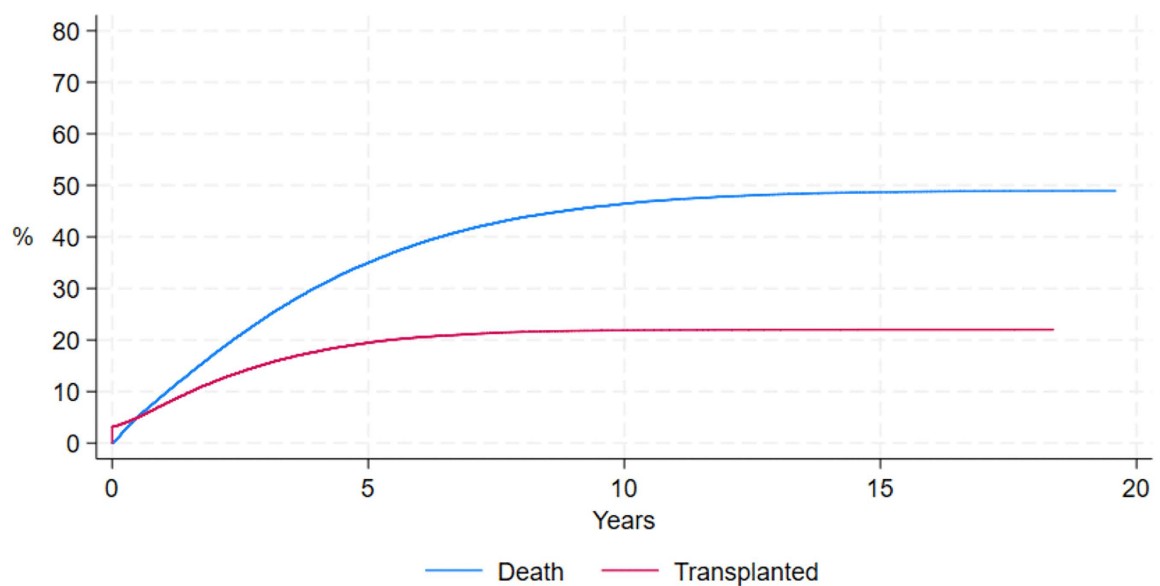

**Fig 1. Time from onset of kidney failure to primary (first) transplant among incident kidney failure patients in Australia and New Zealand (ANZ) (2003–2022), with competing risk of death Top panel: incident youths aged 15-24 years of age. Lower panel: incident adults 25 years and older.** Courtesy of Feruza Kholmurodova, Biostatistician, Australia and New Zealand Dialysis and Transplant Registry (ANZDATA).

## Study procedures

The survey is to be administered as an online, postal, or in-person survey, according to participant preference. Participants will complete the survey independently; however, in rare cases where a participant has a documented cognitive or physical impairment that prevents self-completion, assistance from a parent or caregiver will be permitted. Participants will be asked to complete the survey either within their dialysis facility or at home and given the option of allowing their caring clinical team to access their survey responses. Participants will be given a maximum of 4 weeks to complete the survey. The survey will be conducted electronically using the Qualtrics[XM] Platform™ (Utah, USA) hosted within UoA accredited data centers

An initial version of the survey was created by three authors (KJ, AS, MM) using instruments relevant to the psychosocial challenges faced by young adults and adolescents on dialysis. External experts recommended the inclusion of specific instruments tailored to assess the mental well-being of Māori patients. The survey is structured into eleven sections, each focusing on different aspects of the participants' daily lives, physical health, overall well-being, quality of life, mental health, ethnic identity, self-efficacy, social situation, mood, health literacy, and general health background.

Some sociodemographic and clinical variables will be collected directly from participants via the survey and but supplemented and reconciled with corresponding data collected by site investigators from the ANZDATA Registry, the National (AoNZ) Kidney Transplant Waiting List, and local electronic medical records.

The hard copy surveys and original datasets will be stored in secured, encrypted repository/folders at study sites, and shared with the UoA Research Team as de-identified data extracts labelled only with eID after removal of all personal identifiers.

## Research outcomes and endpoints

The main outcomes in the study are mental well-being and health-related QOL. The former will be assessed from the Short Warwick Short Warwick Edinburgh Mental Well-being Scale (SWEMWBS) [16–21], which evaluates general well-being through a series of statements about feelings and behaviors experienced over the past week. The instrument comprises seven items, and participants provide responses to each on a discrete scale from 0 (never) to 4 (almost always). The items assess feelings of optimism about the future, being useful, relaxation, effective problem-solving, mental clarity, sociability, and firm decision making. The scores for are summed to produce a total score ranging from 0 to 35, with total raw scores transformed into metric ones using the SWEMWBS conversion table [22]. Although there are no local data for AoNZ, scores of 21–27 are typical for healthy people, with lower scores indicating poorer mental wellbeing and higher ones indicating greater mental well-being.

Health-related QOL will be assessed by the EuroQol Group EQ-5D Visual Analogue Scale (VAS) [23], where participants indicate health state on a continuous scale anchored from 0 (the worst possible health state) to 100 (the best possible health state). The population norm for those aged 18–24 years in AoNZ is 82.4 [24].

## Other quantitative variables

**Sociodemographic and clinical variables.** Sociodemographic data will include participant age, gender, educational level, self-declared ethnicity (prioritized according to accepted AoNZ ethnicity data protocols [25–27]), relationship status, current employment, living situation, household income, access to information and communications technology. A general index of socioeconomic status will be calculated using the New Zealand Index of Multiple Deprivation (IMD18), which comprises 29 indicators grouped into seven domains of deprivation: employment, income, crime, housing, health, education and access to services [28]. Rurality of domicile will be classified according to the current generic definitions of Statistics NZ, delineating four urban and eight rural categories [29]. Clinical data will include details of dialysis treatment, kidney transplant listing status, and co-morbidity including the Charlson co-morbidity index [30,31].

**Alternative measures of quality of life.** The primary assessment of health-related QOL will be supplemented by two other measures. In the first, participants will be assessed from their responses in five domains assessed in the EuqoQol Group EQ-5D-3L: mobility, taking care of oneself, usual activities, pain/discomfort, and anxiety/depression [32]. Levels in each of the domains are assessed from questions with Likert response options, ranging from no issues (Level 1) to severe issues (Level 3). The aggregate state for participants is represented by their 5-digit (5D) code (for example, 11111 signifies no issues in any area). The 5D codes will be then transformed into a single utility index using a validated value set for AoNZ [33–35]. The frequency of any reported problems in the AoNZ population using the EQ-5D-3L is 20% for mobility, 4.4% for self-care, 21.5% for usual activities, 40.8% for pain/discomfort, and 21.2% for anxiety/depression [36].

The second alternative measure is from their responses to the Pediatric Quality of Life Inventory 3.0 End-Stage Renal Disease (PedsQL 3.0 ESRD) [37]. Participants will be assessed from their responses on frequency scales in seven domains (1) General Fatigue (4 items), (2) About My Kidney Disease (5 items assessing participants' symptomatic burden from swelling, dizziness, headaches, thirst, and muscle cramps), (3) Treatment Problems (4 items assessing the challenges participants face in adhering to their treatment regimen), (4) Family and Peer Interaction (3 items assessing how treatments impact the participants' relationships with family and friends), (5) Worry (10 items assessing participants' anxieties regarding their medical condition, including concerns about treatment efficacy, surgeries, long-term illness, hospital stays, blood pressure, and infections), (6) Perceived Physical Appearance (3 items assessing participants' feelings related to physical appearance, particularly scars and the side effects of medications), and (7) Communication (5 items assessing how comfortable participants feel when communicating with healthcare providers and explaining their illness to others). The total score will also be assessed as the average of all individual item scores within the scales. Of note, assessments for this instrument use reverse scoring, with 0 (never) = 100, 1 (almost never) = 75, 2 (sometimes) = 50, 3 (often) = 25, and 4 (almost always) = 0.

**Alternative measures of mental well-being.** Mental well-being in those identifying as Māori will be also assessed using the Hua Oranga tool. This scale was initially developed by Sir Mason Durie and Dr Te Kani Kingi and subsequently adapted by Dr Simon Bennett [38–41]. To ensure cultural relevance, alongside generic measures such as EQ-5D and SWEMWBS, we will employ Hua Oranga, a Māori health outcome measure. The Taha scale represents a holistic approach to well-being, featuring four key domains: *"Taha Tinana"* focuses on physical health and self-care, *"Taha Whānau"* emphasizes family relationships and support, *"Taha Hinengaro"* addresses mental and emotional well-being, and *"Taha Wairua"* reflects Māori identity and ancestral connections. Collectively, these domains capture the interconnected aspects of physical, emotional, family, and cultural well-being. The instrument presents 5-point Likert response options to evaluate different domains of mental health and well-being that are important in models of health for indigenous AoNZ peoples. Levels range from 1 (strongly disagree) to 5 (strongly agree), and the scores summed to range between 4–20 for each domain (Table 1)

**Table 1. Taha item scoring for the Hua Oranga Scale.**

| Category | Score Range | Description |
| --- | --- | --- |
| Very Low | 4-8 | Indicates significant challenges in the domain. |
| Low | 9-12 | Reflects some challenges but with opportunities for improvement. |
| High | 13-16 | Represents a good level of performance or wellbeing in the domain. |
| Very High | 17-20 | Highlights strengths and exceptional outcomes in the domain. |

The total score for the Hua Oranga Scale is determined by adding up the scores from all four domains, ranging from 16 to 80. We have incorporated two supplementary questions to further explore Māori identity: "Te reo Māori is important to me in my daily life" and "I feel strongly connected to my culture." These questions are intended to offer a deeper understanding of the individual's connection to their Māori identity and cultural heritage.

**Activities of daily living.** We will explore participants' ability to manage routine tasks in a purely descriptive manner, focusing on how self-sufficient they are in performing these essential activities without assistance from their caregivers. The Activities of Daily Living (ADL) scale for this study uses a binary scoring system, where a score of 1 indicates that the respondents can perform a specific activity, and a score of 0 represents that they cannot. This approach was applied to four types of ADLs, as described in the book "Activities of Daily Living" [42–44]. The categories comprise: basic ADLs. (dressing, eating, toileting, bathing, and mobility); instrumental ADLs (everyday tasks that involve using tools to manage basic responsibilities, like cooking meals, handling phone calls, writing, typing, and managing money); vocational ADLs (activities related to employment or schoolwork); non-vocational ADLs (activities for fun, such as hobbies, leisure activities, and personal interests).

**Physical activity.** Participants' engagement in physical activities will be assessed using the Rapid Assessment of Physical Activity (RAPA) instrument [45,46]. The first section, RAPA 1, includes seven items that inquire about the frequency and intensity of physical activity, requiring binary responses scored as 1 if the respondents can perform at that activity level, and a score of 0 if they cannot. The total score ranges from 1 (infrequent or no physical activity) to 7 (20 + minutes of intense physical activity three or more times weekly). The RAPA 1 scale categorizes overall physical activity into three levels (light, moderate, vigorous). The second section, RAPA 2, asks about attempts to enhance muscle strength and flexibility by engaging in specified exercises.

Although RAPA was originally validated for adults, particularly older adults [47], pilot testing in the current study demonstrated that youth participants could easily understand and respond to RAPA items. To strengthen validity, RAPA responses will be triangulated with the physical functioning subscale of the Pediatric Quality of Life Inventory (PedsQL), allowing cross-verification of self-reported physical activity. Adolescents are advised to engage in at least 60 minutes of moderate to vigorous physical activity daily, including activities that strengthen muscles and bones at least three times per week [47]. Given these guidelines, they might be expected to achieve the maximum RAPA scores, although this is not particularly borne out in available studies. In the largest study using RAPA in healthy youths, 70% of 555 Spanish university students were categorized as sedentary or inactive, 27% moderately active, and 3% active [48]. In contrast, the largest study assessing RAPA in a multinational cohort of dialysis patients with a mean age of 63.4 years showed that 37% of the 5763 people were categorized as sedentary or inactive, 44% as moderately active, and 19% active [49]. There are no studies using the RAPA instrument in AoNZ, in either healthy youths or dialysis patients. The RAPA will therefore be treated as an exploratory measure.

**Self-efficacy.** Self-efficacy will be assessed from the General Self-Efficacy Scale (GSES) instrument [50]. The GSES has 10 items with 4-point Likert response options and evaluates a person's belief in their capability to handle and deal with different circumstances. Scoring is summative with a total score that can vary between 10 and 40. Greater scores indicate higher levels of self-efficacy, showing more confidence in handling challenges and completing tasks. Evidence supporting construct validity of the GSES indicate strong connections with emotions, optimism, and job satisfaction [51]. This implies that people who believe in their capabilities are more likely to have better experiences with positive factors. The GSES is inversely related to depression, stress, and anxiety, showing that increased self-efficacy is linked to reduced levels of these adverse mental conditions.

**Social anxiety.** We will explore this construct using the DSM-5 Social Anxiety Disorder Severity Scale (SAD-D) [52]. In this scale, each item is scored using a discrete 5-point frequency scale ranging from 0 (never) and 4 (all the time). The total score is summative and ranges between 0–40, with higher scores indicating more severe anxiety. A variety of cut-off values have been suggested, and exploration of this recently developed scale is ongoing. It has been reported that individuals seeking help for social anxiety had a mean SAD-D score of 25.7 [53], although a clinically useful cut-off for assessments has recently been suggested at 19 [54].

**Depression.** Depressive symptoms will be assessed using the Center for Epidemiological Studies Depression Scale for Children (CES-DC). The instrument comprises 20 items using a 4-point Likert scale, with scores ranging 0 (not at

all) to 3 (a lot). Reverse scoring is required for some positively framed items. The total score is summative and ranges between 0–60, with higher scores indicating increased chance of depression. A well accepted cut-off has been established by the developers, namely Weissman et al, whereby a score >15 suggests presence of depressive symptoms in a young adult requiring additional evaluation [55]. Data in dialysis populations suggest that about quarter to half of people screened using this tool are likely to have results indicating depression [56–59].

**Health literacy.** We will assess health literacy using a three-item instrument. The single items within the instrument have each been separately validated against definitive instruments [60,61], and the combination used as a single measure of health literacy [62,63]. Reversing the first item, scores of nine or above on the combined scale indicate suboptimal health literacy. Of note, this scale only assesses the ability of patients to locate, interpret and apply health information (classically defined health literacy) [64], rather than the extent to which health systems provide accessible comprehensible information (so-called health information fluency) [65,66].

**Open-ended questions.** Free text comments will be sought from participants around situations causing social anxiety and other factors of general concern and these will be analyzed using both inductive and deductive qualitative content analysis [67].The qualitative data will be coded and analyzed by two members of the research team and any discrepancies in coding with be resolved via consensus.

## Data analysis

We will use structural equations models (SEM) to explore possible relationship between physical and psychometric predictors and the outcomes of interest, namely mental wellbeing and health related quality of life. There are no existing conceptual models linking the psychosocial factors in this study to the outcomes, and exploratory rather than confirmatory modelling will be performed using variance-based SEM [68–71]. We chose this analytical technique over traditional covariance-based structural equation modelling, since it is more often used for prediction and theory development, as opposed to the latter which is more used for theory testing and confirmation. In variance-based SEM, the psychometric variables in our survey will be used to generate weighted composites, and paths estimated between the composites in a manner broadly similar to multiple linear regression. Because of the likely small sample size and the exploratory nature of the modelling, we will not perform rigorous assessments of items within the instruments for internal validity through confirmatory measurement models, and scores from the survey items will be used as they were originally published.

For each of the two outcomes of interest, the following will be explored as mediating or exogenous variables: depression as defined by the CES-DC scale, self-efficacy defined as defined by the GSE scale, social phobia as defined by the SAD-D scale, physical activity defined as defined by the RAPA scale, and the constructs from the PedsQL 3.0 QOL End Stage Renal Disease Module, namely general fatigue, kidney disease symptoms, treatment problems, family and peer Interactions, worry about medical care, perceived physical appearance, and communication with medical/ nursing/ allied staff. Simple correlational analysis will be used to identify the key scales associating either positively or negatively with each of the two outcomes of interest, and these will be used by the investigators to create plausible structural relationships, derived from both the literature and their own cumulative clinical experience. Final model structures will be dictated by the aims and objectives described above, and refined according to model fit, face-validity in a clinical sense, and the presence and extent of support for a given model structure from other published research.

Depending on the pattern of survey responses, we may proceed to cluster analysis and descriptive profiling as secondary post hoc activities, with or without latent class analyses looking for subgroups that we cannot cluster using manifest variables. The benefit of this approach will arise from determine demographic and behavioral variables differ across the clusters, which might in turn create further actionable insights within discrete respondent groups of our population. For example, the difference in self-efficacy and other characteristics can be explored in clusters defined by dependent and independent dialysis to look for levers to improve self-efficacy or improve uptake of home dialysis, depending on the respondent results.

## Discussion

At the current time, most physical and psychological initiatives in this group focus on medical rehabilitation and generic coping acceptance strategies, respectively. In general, there is less emphasis on specific interventions that focus on developing youth physical and social skills using a modern educational science lens. Such interventions might create safe, positive problem-solving skills for young people, especially if learning is embedded in realistic contexts with authentic tasks [72,73].

The challenges encountered by youth on dialysis in AoNZ and elsewhere are significant and different from their adult counterparts. In AoNZ, structural racism and health inequities within the system disproportionately disadvantage Māori and Pacific youths, contributing to challenges in quality of life, psychological distress, and increased social limitations compared to their European counterparts [74,75]. This systemic bias is evident in the fact that 50% of the 154 youths on dialysis in AoNZ between 1 January 2000 and 31 December 2014 were from these ethnic groups.

We expect our study to cast light on the presence and extent of physical and psychosocial needs of youth on dialysis and suggest specific areas for intervention.

## Author contributions

**Conceptualization:** Kavitha Jaganathan, Anna Serlachius, Chanel Prestidge, Fredric S. Doss, Mark Marshall.

**Data curation:** Kavitha Jaganathan, Anna Serlachius, Chanel Prestidge, Fredric S. Doss, Mark Marshall.

**Formal analysis:** Kavitha Jaganathan, Anna Serlachius, Mark Marshall.

**Funding acquisition:** Kavitha Jaganathan, Anna Serlachius, Mark Marshall.

**Investigation:** Kavitha Jaganathan, Chanel Prestidge, Fredric S. Doss, Mark Marshall.

**Methodology:** Kavitha Jaganathan, Anna Serlachius, Chanel Prestidge, Fredric S. Doss, Mark Marshall.

**Project administration:** Kavitha Jaganathan, Anna Serlachius.

**Resources:** Kavitha Jaganathan, Anna Serlachius, Chanel Prestidge, Fredric S. Doss, Mark Marshall.

**Software:** Kavitha Jaganathan, Anna Serlachius, Mark Marshall.

**Supervision:** Anna Serlachius, Mark Marshall.

**Validation:** Kavitha Jaganathan, Chanel Prestidge, Fredric S. Doss, Mark Marshall.

**Writing – original draft:** Kavitha Jaganathan, Anna Serlachius, Mark Marshall.

**Writing – review & editing:** Kavitha Jaganathan, Anna Serlachius, Chanel Prestidge, Fredric S. Doss, Mark Marshall.

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
