## [Decision Letter · Decision Letter 0]

26 Nov 2025

Dear Dr. Marshall,

Thank you for submitting your manuscript to PLOS ONE. After careful consideration, we feel that it has merit but does not fully meet PLOS ONE’s publication criteria as it currently stands. Therefore, we invite you to submit a revised version of the manuscript that addresses the points raised during the review process.

We look forward to receiving your revised manuscript.

Kind regards,

Hansani Madushika Abeywickrama, Ph.D.

Academic Editor

PLOS ONE

Journal Requirements:

“University of Auckland Open Access Grant as part of general PhD funding”

4. Please include captions for your Supporting Information files at the end of your manuscript, and update any in-text citations to match accordingly. Please see our Supporting Information guidelines for more information: http://journals.plos.org/plosone/s/supporting-information .

Additional Editor Comments :

Thank you for submitting this important protocol examining the health and psychosocial outcomes of youths receiving dialysis. Both reviewers agree that the study addresses a notable gap and has the potential to make a meaningful contribution to nephrology research. Please revise the manuscript addressing the points raised by reviewers. Ensure clarity, consistency, and methodological appropriateness throughout the updated protocol. We look forward to receiving a carefully revised version.

Reviewers' comments:

Reviewer's Responses to Questions

**Comments to the Author**

1. Does the manuscript provide a valid rationale for the proposed study, with clearly identified and justified research questions?

Reviewer #1: Yes

Reviewer #2: Yes

2. Is the protocol technically sound and planned in a manner that will lead to a meaningful outcome and allow testing the stated hypotheses?

Reviewer #1: Yes

Reviewer #2: Yes

3. Is the methodology feasible and described in sufficient detail to allow the work to be replicable?

Reviewer #1: Yes

Reviewer #2: Yes

4. Have the authors described where all data underlying the findings will be made available when the study is complete?

Reviewer #1: Yes

Reviewer #2: Yes

5. Is the manuscript presented in an intelligible fashion and written in standard English?

Reviewer #1: Yes

Reviewer #2: Yes

You may also provide optional suggestions and comments to authors that they might find helpful in planning their study.

Reviewer #1: This study addresses a vital gap in the care of youths on dialysis. The data will be highly relevant to the nephrology community. However, the current statistical plan (SEM) is a significant weakness given the limited patient population. Shifting the analytical focus toward clustering and descriptive profiling would make the protocol scientifically robust.

Nevertheless, there is a real need and benefit to collecting comprehensive structured data within this unstudied population and PLOS ONE should support the proposal.

Note to the Editor: If this methodology paper is accepted, PLOS ONE should be the venue for the results of the study to ensure the complete research cycle is available in one place

Reviewer #2: Thank you for submitting this protocol for a study in an important population

A few comments

The word 'youth' throughout sometimes needs to be plural wonder if young adult or young adults might be better

The PPIE section there is sentence started and not completed

Just some clarity on exclusion criteria as further on you state has a 'documented cognitive or physical impairment that prevents self-completion, assistance from a parent or caregiver will be permitted'- just need to be clear why is this different to what is described as severe in the exclusion criteria

An extensive list of outcome measures including clinical variables just a question as this is for people on dialysis have you considered any questions on their role within their dialysis treatment relating to self efficacy?

**Do you want your identity to be public for this peer review?** For information about this choice, including consent withdrawal, please see our Privacy Policy

Reviewer #1: No

Reviewer #2: No

---

## [Author Response · Author response to Decision Letter 1]

26 Nov 2025

Hansani Madushika Abeywickrama, Ph.D.

Academic Editor

PLOS ONE

Dear Dr Abeywickrama

Response to Reviewers

Journal Requirements:

Thank you, we have made changes in style to comply with requirements.

“University of Auckland Open Access Grant as part of general PhD funding”

Thanks you we have done this.

Thank you. We apricate that the ambiguity in "reasonable request" has been criticized in open science discussions, and that many advocate for clearer data sharing commitments like public repository deposits to avoid gatekeeping. We have clarified the restrictions on data sharing, which we have expanded upon on page 17 of the manuscript. Of note, some of these restrictions are legal. See below:

#############################START##############################

Authors agree to make data and materials supporting the results or analyses presented in their paper available upon reasonable request. It is up to the authors to determine whether a request is reasonable. Several factors will be considered in evaluating requests:

• Legitimacy of scientific purpose: requests will be viewed more favorably if the intention is replication, meta-analysis, secondary analysis, or educational use.

• Requester credentials: requests from researchers at recognized institutions with relevant expertise will be viewed more favorably.

• Feasibility and burden: reasonable requests should not impose excessive work on the authors

• Ethical and legal constraints: certain data containing sensitive information have restrictions on sharing, as specified in data use agreements and ethics / IRB approvals. In particular, as a matter of national policy in Aotearoa New Zealand, there is a relative prohibition around sending Māori research data overseas consistent with the principles of Māori self-determination and control over data about their communities, rooted in Te Tiriti o Waitangi (the Treaty of Waitangi) and as specified in the (NZ) Data and Statistics Act 2022. Sharing of such data would require an evaluation from the national (NZ) Health and Disability Ethics Committee and as well as IRBs of all the participating hospitals, who will decide whether it is possible enforce NZ rules and norms pertaining to such data, including but not limited to: (1) whether the overseas jurisdictions have comparable data protection safeguards to NZ; (2) whether there will be culturally appropriate representation on overseas governance committees to ensure that the data are handled in culturally appropriate ways with a good understanding of the NZ Māori community that the data relate to

• Specificity: only specific requests for variables relevant to well-defined research question will be considered

• Timing: requests made shortly after publication will be considered whereas those made a substantial amount of time later may not be possible when authors have moved on or data storage might have lapsed.

#############################FINISH##############################

N/A

N/A

Thank you, and checked

Reviewer #1comments:

The current statistical plan (SEM) is a significant weakness given the limited patient population. Shifting the analytical focus toward clustering and descriptive profiling would make the protocol scientifically robust.

Thank you, this is a very useful and helpful comment.

Firstly, we think a structural / causal analysis is warranted to create actionable insights with a likely benefit in idealistically a deterministic manner (but of course a practically stochastic one). We have been clear that we are using PLS-SEM rather than CB-SEM. The former is very forgiving to smaller datasets and would allow reasonable statistical inference in this sample, in our estimation. The corresponding author has several publications using this methodology in samples with n < 100, and is confident of being able to obtain at least a signal of findings if they are there to be found. The overall chance of model convergence and statistical significance is hampered by the small sample size, of course, but this leads to Type II rather than Type I errors. The group is happy to work within this constraint.

The suggestion around cluster analysis +/- hierarchical clustering with dendrograms +/- decision tree methods like CHAID or CART etc is a great idea, although we did not specify this analysis in our statistical analysis plan, let alone details such as the number of clusters in advance. This will be a great post hoc study and we might include latent class analysis as well, in case there are subgroups that we cannot identify a priori, but which become clear based on patterns of survey responses. Once we have those clusters mapped, we will determine how demographic and behavioral responses differ across them. Such an approach might create further actionable insights within discrete respondent groups in our sample.

We have added a statement of intent around these extended analyses in the discussion on Page 15.

Reviewer #1comments:

The word 'youth' throughout sometimes needs to be plural wonder if young adult or young adults might be better

Thank you. We too have struggled with this at times. “Youth" can be a singular noun, referring to one young person. It can also be used collectively for a group of young people, as an uncountable noun. We have gone through the paper and made sure that “youth” is used as a singular noun or used collectively, and made sure that we have used the term “youths” for more than one person who all happen to fit the conceptual definition.

Speaking of which, "youth" is an official demographic term. For statistical and research purposes, the UN defines 'youth' as persons between ages 15 and 24, a definition established in 1981 during preparations for the International Youth Year. All UN statistics on youth are based on this definition, as reflected in annual yearbooks on demography, education, employment, and health United Nations. WHO defines adolescents as individuals between 10 and 19 years, young people as those aged 10-24, and youth as those aged 15-24.

We feel that we should be clear about which definition of younger persons we are studying, and introducing the term “young adults” we think would create confusion.

The PPIE section there is sentence started and not completed

Thank you, this error has been corrected.

Just some clarity on exclusion criteria as further on you state has a 'documented cognitive or physical impairment that prevents self-completion, assistance from a parent or caregiver will be permitted'- just need to be clear why is this different to what is described as severe in the exclusion criteria

Thank you for this. The issue is not really one that pertains to communication difficulties, but rather pertains to those patients with severe functional difficulties or disabilities that are not appropriate for assessment by the survey instrument. The verbiage in the paper was carried through from the protocol and was clearly an error. We have corrected this in the manuscript on page 7.

An extensive list of outcome measures including clinical variables just a question as this is for people on dialysis have you considered any questions on their role within their dialysis treatment relating to self efficacy?

Thank you for this great suggestion. We had not though of this, and will triangulate and correlate GSES with dependent versus independent modalities of dialysis. This maybe an additional paper and would suit a clustering and descriptive profiling paper which we should do as a post-hoc exercise. We have added statement of intent around these extended analyses in the discussion on Page 15-16.

---

## [Decision Letter · Decision Letter 1]

15 Dec 2025

Protocol for the ‘Exploration of Mental Well-Being, Health-Related Quality of Life, and Psychosocial Outcomes in Youths on Dialysis in New Zealand: A Cross-Sectional Methods Study’

PONE-D-25-52324R1

Dear Dr. Marshall,

We’re pleased to inform you that your manuscript has been judged scientifically suitable for publication and will be formally accepted for publication once it meets all outstanding technical requirements.

Kind regards,

Hansani Madushika Abeywickrama, Ph.D.

Academic Editor

PLOS One

Additional Editor Comments (optional):

Reviewers' comments:

Reviewer's Responses to Questions

**Comments to the Author**

1. Does the manuscript provide a valid rationale for the proposed study, with clearly identified and justified research questions?

Reviewer #1: Yes

Reviewer #2: Yes

2. Is the protocol technically sound and planned in a manner that will lead to a meaningful outcome and allow testing the stated hypotheses?

Reviewer #1: Yes

Reviewer #2: Yes

3. Is the methodology feasible and described in sufficient detail to allow the work to be replicable?

Reviewer #1: Yes

Reviewer #2: Yes

4. Have the authors described where all data underlying the findings will be made available when the study is complete?

Reviewer #1: Yes

Reviewer #2: Yes

5. Is the manuscript presented in an intelligible fashion and written in standard English?

Reviewer #1: Yes

Reviewer #2: Yes

You may also provide optional suggestions and comments to authors that they might find helpful in planning their study.

Reviewer #1: The authors have addressed the questions raised. They acknowledge the limitations of their project, which, in my opinion, are outweighed by the collective interest of the approach for the discipline. This article could therefore be relevant for publication in PLOS.

Reviewer #2: Thank you for taking time to address the comments I have no further comments to add, good luck with the study

**Do you want your identity to be public for this peer review?** For information about this choice, including consent withdrawal, please see our Privacy Policy

Reviewer #1: No

Reviewer #2: No

---

## [Editor Report · Acceptance letter]

PONE-D-25-52324R1

PLOS One

Dear Dr. Marshall,

I'm pleased to inform you that your manuscript has been deemed suitable for publication in PLOS One. Congratulations! Your manuscript is now being handed over to our production team.

Kind regards,

on behalf of

Dr. Hansani Madushika Abeywickrama

Academic Editor

PLOS One